EMBO
Molecular Medicine

# BNT162b2 vaccine induces antibody release in saliva: a possible role for mucosal viral protection?

Abbass Darwich[1],[†], Chiara Pozzi[2],[†] (ID), Giulia Fornasa[2] (ID), Michela Lizier[2] (ID), Elena Azzolini[1,2], Ilaria Spadoni[1], Francesco Carli[3], Antonio Voza[1,2], Antonio Desai[1,2] (ID), Carlo Ferrero[2] (ID), Luca Germagnoli[2], ICH COVID-19 Task-force[‡], Alberto Mantovani[1,2,4] & Maria Rescigno[1,2],* (ID)

## Abstract

Vaccination against an airborne pathogen is very effective if it induces also the development of mucosal antibodies that can protect against infection. The mRNA-based vaccine-encoding SARS-CoV-2 full-length spike protein (BNT162b2, Pfizer/BioNTech) protects also against infection despite being administered systemically. Here, we show that upon vaccination, cognate IgG molecules are also found in the saliva and are more abundant in SARS-CoV-2 previously exposed subjects, paralleling the development of plasma IgG. The antibodies titer declines at 3 months from vaccination. We identified a concentration of specific IgG in the plasma above which the relevant IgG can be detected in the saliva. Regarding IgA antibodies, we found only protease-susceptible IgA1 antibodies in plasma while they were present at very low levels in the saliva over the course of vaccination of SARS-CoV-2-naïve subjects. Thus, in response to BNT162b2 vaccine, plasma IgG can permeate into mucosal sites and participate in viral protection. It is not clear why IgA1 are detected in low amount, they may be proteolytically cleaved.

**Keywords** BNT162b2; IgA; IgG; mucosal immunity; SARS-CoV-2
**Subject Categories** Immunology; Microbiology, Virology & Host Pathogen Interaction

## Introduction

The mRNA-based vaccine-encoding SARS-CoV-2 full-length spike protein (BNT162b2, Pfizer/BioNTech) induces a strong neutralizing antibody response already after two vaccine doses (Walsh *et al*, 2020). Recently, the results of a "real-world" vaccination trial in Israel (Dagan *et al*, 2021) and an interim study after one dose administration in the United Kingdom (Vasileiou *et al*, 2021) have been published. The authors have shown that besides protecting from disease, the vaccine has a strong effect on SARS-CoV-2 infection. The Pfizer/BioNTech vaccine efficacy against the WT virus form ranges from 46% at 14–20 days from the first dose, 60% at 21–27 days from the first dose, and 92% at 7 days from the second dose (Dagan *et al*, 2021). Moreover, although there is a substantial reduction in efficacy, this vaccine has proven effective against infection and hospitalization due to the delta variant (Risk *et al*, 2022) and against hospital admission for COVID-19 caused by the omicron variant (Collie *et al*, 2022).

How can this protection from infection be explained? One would expect that a systemic vaccine induces a strong systemic IgG/IgA response that protects only once the virus has entered the host, but the finding that infection is strongly reduced in vaccinated people suggests that the immunoglobulins have reached mucosal surfaces for barrier protection. In the saliva, differently from other mucosal sites, both IgA and IgG can be detected. IgA are the classical mucosal neutralizing antibodies, which are normally found in secreted fluids in a dimeric form bound to the secretory component that is acquired from the poly Ig receptor after translocation across epithelial cells (Cerutti & Rescigno, 2008). Salivary IgAs are secreted from mucosal tissues of the tonsils and adenoids and thanks to their secretory components that are protected from degradation (Brandtzaeg, 2007). During natural SARS-CoV-2 infection IgG, IgM, and IgA are all found not only in the serum (Pisanic *et al*, 2020; Wang *et al*, 2021), but also in saliva and bronchoalveolar fluids shortly after symptoms onset (Isho *et al*, 2020; Sterlin *et al*, 2021). Recently, it has been shown that also mRNA-based SARS-CoV-2 vaccination induces a weak mucosal response, and SARS-CoV-2-specific IgG and IgA are detected in the saliva and nasal fluid (Chan *et al*, 2021; Guerrieri *et al*, 2021; Klingler *et al*, 2021; Azzi *et al*, 2022). However, the duration of saliva immunoglobulins and the nature of the IgA have not been reported. The saliva is a site were the virus is detected for a long time, and the oral mucosal epithelium may contribute to viral infection (To *et al*, 2020; Ali & Sweeney, 2021; Kwon

1   Department of Biomedical Sciences, Humanitas University, Milan, Italy
2   IRCCS Humanitas Research Hospital, Milan, Italy
3   Department of Informatics, Università degli Studi di Torino, Torino, Italy
4   The William Harvey Research Institute, Queen Mary University of London, London, UK
    *Corresponding author. E-mail: maria.rescigno@hunimed.eu
    †These authors contributed equally to the work
    ‡ICH COVID-19 task-force members given in Appendix.

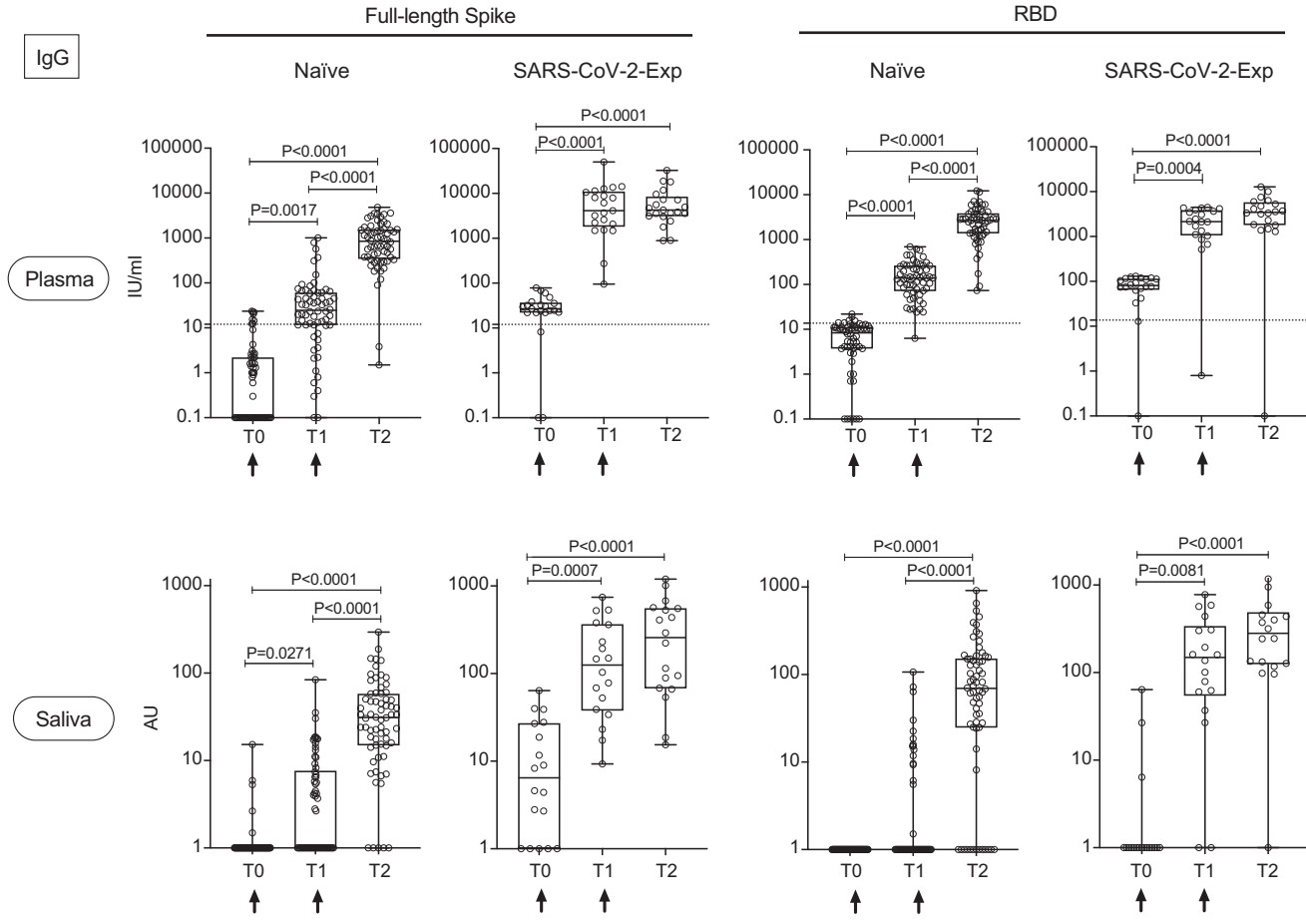

**Figure 1. BNT162b2 vaccination induces the release of SARS-CoV-2-specific IgG in plasma and saliva.**

Antibody response in plasma and saliva at different time points (T0, T1, T2) from BNT162b2 vaccination in naïve (plasma: $n = 64$; saliva: $n = 66$) and SARS-CoV-2-Exp (plasma: $n = 21$; saliva: $n = 18$) subjects. Subjects received the first vaccine dose at day 0 (T0) and the 2nd dose at day 21 (T1) (indicated with an arrow). T2 corresponds to 7–10 days after the 2nd dose. Plasma and saliva were tested for IgG to full-length spike and its receptor-binding domain (RBD). For plasma samples the titers of antigen-specific Ig are expressed in IU/ml. LoD is indicated by a dotted line: LoD (spike) = 12, LoD (RBD) = 13.8. For saliva samples, the titers of antigen-specific Ig were normalized by dividing the values of SARS-CoV-2-specific Ig by total IgG concentration of each sample. The normalization was applied only to values higher than LoD. The adjusted values are expressed in AU. The box plots show the interquartile range, the horizontal lines show the median values, and the whiskers indicate the minimum-to-maximum range. Each dot corresponds to an individual subject. Log scale on y axis. P-values were determined using the Friedman test with the Dunnett's multiple comparison test.

Source data are available online for this figure.

et al, 2021). Hence, mucosal antibodies, and particularly secreted neutralizing IgA, patrol the nasopharyngeal and oral mucosa contributing to protecting against infection (Isho *et al*, 2020; Sterlin *et al*, 2021). Salivary IgGs derive mostly from the microvasculature of gingival crevices, and hence from the serum (Brandtzaeg, 2007). As both IgG and IgA serum antibodies are quickly produced upon BNT162b2 vaccination (Samanovic *et al*, 2021), here, we characterized the SARS-CoV-2-specific IgG, IgA, IgA1 and IgA2 response in the plasma and saliva of COVID-19 patients and vaccinated individuals, both naïve and previously exposed to SARS-CoV-2.

## Results and Discussion

We analyzed the development of the IgG and IgA antibody response to SARS-Cov2 full-length spike protein or its receptor-binding domain (RBD), versus nucleocapsid N protein in the plasma and saliva of 92 subjects immunized with BNT162b2 (71 naïve and 21 previously exposed to SARS-CoV-2). We analyzed the antibody titers at baseline [the time of the first dose (T0)], at the time of the second dose (21 days after the first, T1) and 7–10 days after the second dose (T2). We showed that IgGs to both spike and RBD steadily increased in the plasma and saliva of SARS-CoV-2-naïve subjects after the first and second vaccine doses (Fig 1). Interestingly, we found that similar to the plasma, the IgG response in the saliva of SARS-CoV-2 previously exposed subjects (SARS-CoV-2-Exp) was already maximal at the time of the second dose (T1) (Fig 1). This is in agreement with what we and others previously published on plasma IgG, showing that one dose of the vaccine is sufficient to induce a very strong antibody response in SARS-CoV-2 previously exposed individuals (Abu Jabal *et al*, 2021; Krammer *et al*, 2021; preprint: Levi *et al*, 2021; Saadat *et al*,

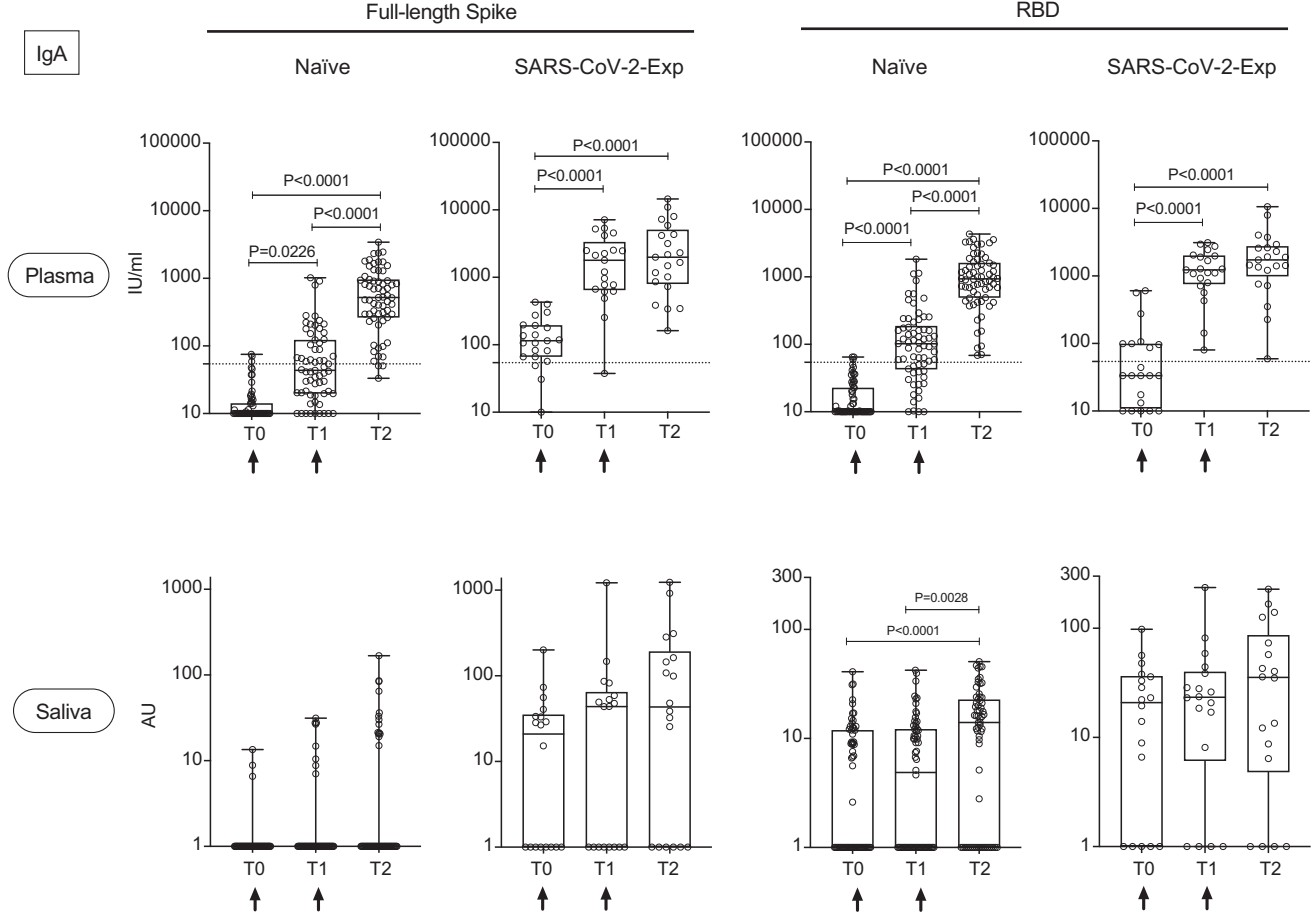

**Figure 2. BNT162b2 vaccination induces the release of SARS-CoV-2-specific IgA in plasma and saliva.**

Antibody response in plasma and saliva at different time points (T0, T1, T2) from BNT162b2 vaccination in naïve (*n* = 66) and SARS-CoV-2-Exp (plasma: *n* = 21; saliva: *n* = 18) subjects. Subjects received the first vaccine dose at day 0 (T0), and the 2nd dose at day 21 (T1) (indicated with an arrow). T2 corresponds to 7–10 days after the 2nd dose. Plasma and saliva were tested for IgA to full-length spike and its receptor-binding domain (RBD). For plasma samples the titers of antigen-specific Ig are expressed in IU/ml. LoD is indicated by a dotted line: LoD (spike) = 54.22, LoD (RBD) = 54.08. For saliva samples, the titers of antigen-specific Ig were normalized by dividing the values of SARS-CoV-2-specific Ig by total IgA concentration of each sample. The normalization was applied only to values higher than LoD. The adjusted values are expressed in AU. The box plots show the interquartile range, the horizontal lines show the median values, and the whiskers indicate the minimum-to-maximum range. Each dot corresponds to an individual subject. Log scale on *y* axis. *P*-values were determined using the Friedman test with the Dunnett's multiple comparison test.

Source data are available online for this figure.

2021). However, this was not true for the SARS-CoV-2-specific IgA response. As shown in Fig 2, the IgA response followed a similar pattern to IgG in the plasma both in naïve and SARS-CoV-2-Exp subjects. By contrast, the anti-spike IgA response was very low in the saliva of vaccinated naïve subjects at any time points, while the anti-RBD IgA response increased significantly only after the second dose (T2) compared with T0 and T1 (Fig 2). The IgA titers in the saliva of SARS-CoV-2-Exp individuals was higher compared to that in the vaccinated naïve subjects, and they remained constant over time, without reaching any statistical difference. Interestingly, we found a correlation between the amount of spike SARS-CoV-2-specific IgG in plasma versus that in the saliva. However, the data would not completely fit a linear regression curve (Fig EV1A) with a spearman correlation of 0.4 and would highlight a threshold of plasma SARS-CoV-2-specific IgG over which it was possible to detect the relevant IgG also in the saliva. The area under the

receiver-operating characteristic (ROC) curve (0.94 ± 0.03, 95% CI 0.8865 to 0.9883, *P* < 0.0001) confirmed a good correlation of the anti-spike IgG plasma and saliva levels (Fig EV1B). In order to estimate the threshold point, we computed the entire ROC curve obtaining combinations of specificity and sensitivity metrics for different cut-off values. To select the optimal threshold value, we computed the F1 score (Baeza-Yates & Ribeiro-Neto, 1999) and selected the value that maximized this metric via a machine-learning approach (see methods). Using this tool, we selected as optimal threshold of 1.893 (log plasma) that corresponds to 78.16 IU/ml, which featured an F1 score of 92.8%, a sensitivity 100%, and a specificity of 86.5% (Fig EV1C and Appendix Table S1).

We then compared the magnitude of the IgG and IgA response of vaccinated individuals (either naïve or previously exposed to SARS-CoV-2) versus that of COVID-19 patients during natural infection. It was clear that both plasma anti-spike and anti-RBD IgG and

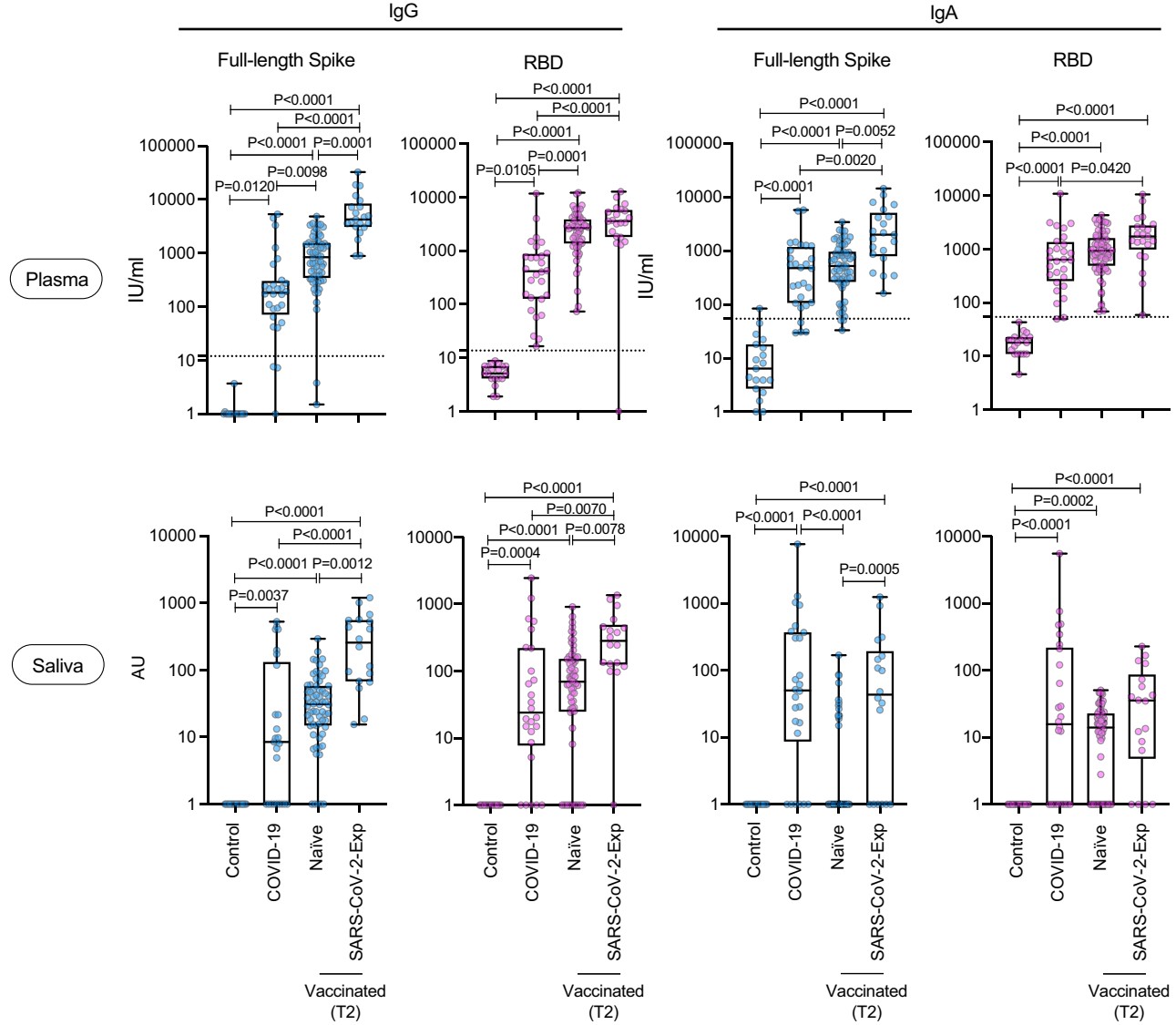

**Figure 3. Release of SARS-CoV-2-specific IgG and IgA in plasma and saliva after BNT162b2 vaccination and in COVID-19 patients.**

The antibody response of vaccinated naïve (plasma: n = 64; saliva n = 66) or SARS-CoV-2-Exp (plasma: n = 21; saliva: n = 18) subjects at 7–10 days after the second dose (T2) was compared to that obtained in COVID-19 patients (plasma n = 28; saliva n = 26). As a control, SARS-CoV-2-specific antibody response was analyzed in nonvaccinated subjects (n = 19, Control). Plasma and saliva were tested for IgA to full-length spike and its receptor-binding domain (RBD). For plasma samples, the titers of antigen-specific Ig are expressed in IU/ml. LoD is indicated by a dotted line: LoD (spike IgG) = 12, LoD (RBD IgG) = 13.8, LoD (spike IgA) = 54.22, LoD (RBD IgA) = 54.08. For saliva samples, the titers of antigen-specific Ig were normalized by dividing the values of SARS-CoV-2-specific Ig by total IgA or total IgG concentrations of each sample. The normalization was applied only to values higher than LoD. The adjusted values are expressed in AU. The box plots show the interquartile range, the horizontal lines show the median values, and the whiskers indicate the minimum-to-maximum range. Each dot corresponds to an individual subject. Log scale on y axis. P-values were determined using the Kruskal–Wallis test with the Dunn's multiple comparison test.

Source data are available online for this figure.

IgA were equally found in the three groups but were statistically significantly higher in SARS-CoV-2-Exp-vaccinated subjects than in COVID-19 patients (Fig 3). A similar scenario was observed for saliva IgG both anti-RBD and spike (Fig 3). By contrast, the amount of anti-spike IgA in the saliva of SARS-CoV-2-Exp individuals and COVID-19 patients was similarly higher compared with nonvaccinated (control group) and vaccinated naïve subjects. Finally, anti-RBD IgA in the saliva was similarly higher in vaccinated people and COVID-19 patients compared to that in the control group, regardless

of previous exposure to SARS-CoV-2 (Fig 3). As expected, we found high levels of IgG and IgA to nucleocapsid N protein (not present in the vaccine) both in plasma and saliva of COVID-19 patients (Fig EV2). Accordingly, also vaccinated subjects previously exposed to SARS-CoV-2 virus showed intermediate levels of IgG anti-N protein, but only in the plasma (Fig EV2).

These data strongly support the notion that antibodies from the plasma permeate the saliva where serum antibodies can filter via the microvasculature of gingival crevices (Brandtzaeg, 2007).

Human IgA is present in two isotypes: IgA1 which are more abundant in the serum (80%) and IgA2 (Macpherson *et al*, 2008). IgA2 are more resistant to proteases as they lack a 13 aminoacid sequence in the hinge region containing a protease cleavage site and are the ones found primarily in the secreted fluids (Macpherson *et al*, 2008). Pneumococcal vaccination induces the development also of IgA2 antibodies (Lue *et al*, 1988). Thus, to evaluate whether

the low levels of salivary IgA were related to their serum origin, which would make them particularly susceptible to proteases, we evaluated which was the isotype of IgA in both plasma and saliva. As shown in Fig 4A, anti-spike IgA in the plasma is of IgA1 isotype. Indeed, as expected, the IgA2 isotype was not present in the plasma. In the saliva of vaccinated naïve subjects, we observed only a very low amount of the IgA1 isotype, while in vaccinated SARS-CoV-2-Exp

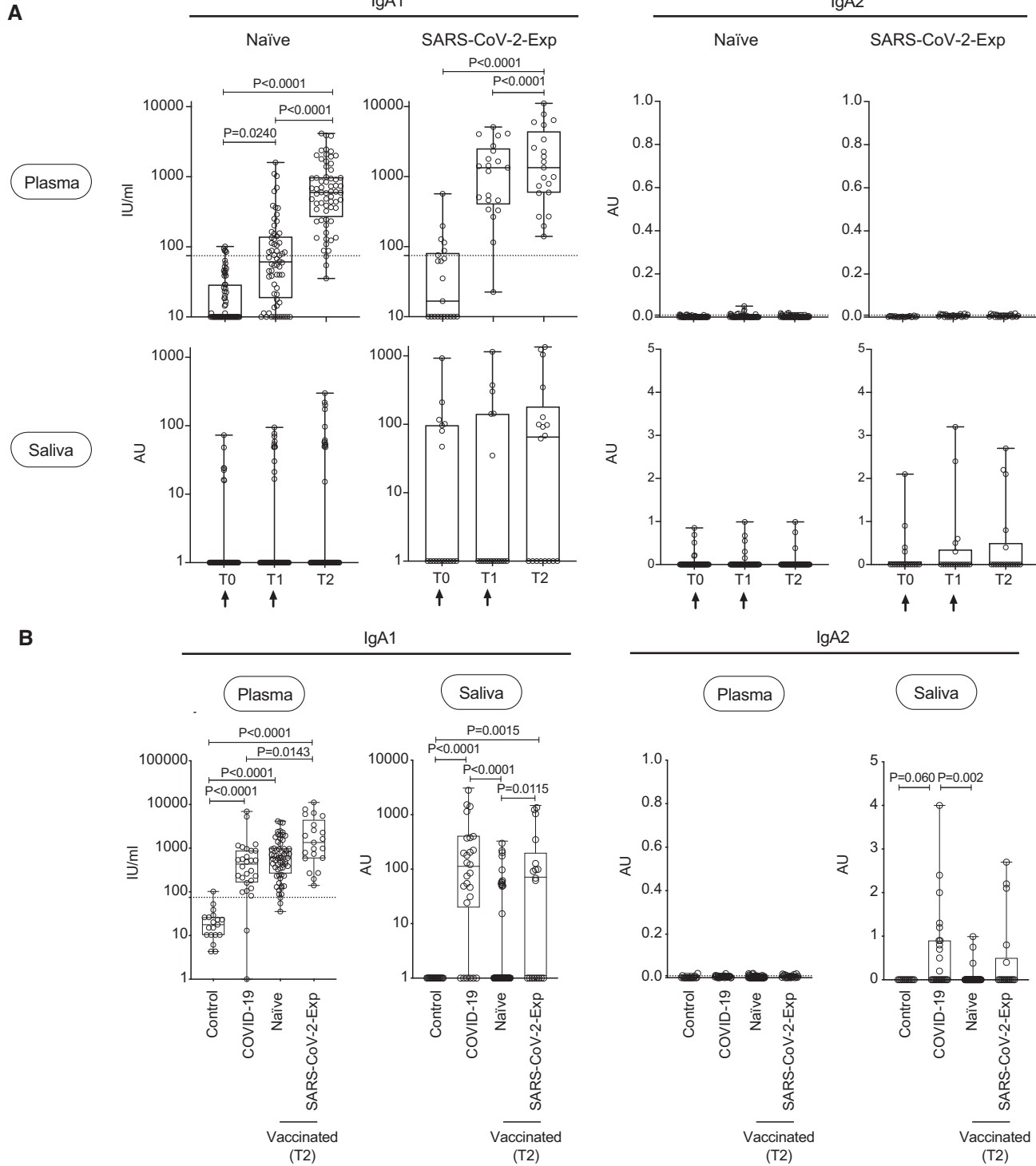

**Figure 4.**

◄

**Figure 4.   Release of SARS-CoV-2-specific IgA1 and IgA2 antibodies in plasma and saliva after BNT162b2 vaccination and in COVID-19 patients.**

A   IgA1 and IgA2 subtypes specific for spike protein were measured in plasma and saliva at different time point (T0,T1, T2) of vaccinated naïve (plasma: $n = 64$; saliva: $n = 63$ (IgA1), $n = 55$ (IgA2)) or SARS-CoV-2-Exp (plasma: $n = 21$; saliva: $n = 18$) subjects. Subjects received the first vaccine dose at day 0 (T0), and the 2nd dose at day 21 (T1) (indicated with an arrow). T2 corresponds to 7–10 days after the 2nd dose. For plasma samples, the titers of antigen-specific IgA1 and IgA2 are expressed in IU/ml and AU, respectively (see Methods). LoD is indicated by a dotted line: LoD (IgA1) = 74.64; LoD (IgA2) = 0.009. For saliva samples, the titers of antigen-specific Ig were normalized by dividing the values of SARS-CoV-2-specific Ig by total IgA concentration of each sample. The normalization was applied only to values higher than LoD. The adjusted values are expressed in AU.

B   IgA1 and IgA2 subtypes specific for spike protein were measured in plasma and saliva of vaccinated naïve and SARS-CoV-2-Exp subjects 7–10 days after second dose (T2), and the antibody response was compared to that obtained in COVID-19 patients (plasma: $n = 28$; saliva: $n = 26$) and in nonvaccinated subjects ($n = 19$, Control).

Data information: The box plots show the interquartile range, the horizontal lines show the median values, and the whiskers indicate the minimum-to-maximum range. Each dot corresponds to an individual subject. Log scale on $y$ axis (only for IgA1). $P$-values were determined using the Friedman test with the Dunnett's multiple comparison test (A) or the Kruskal–Wallis test with the Dunn's multiple comparison test (B).

Source data are available online for this figure.

individuals, we found a higher basal level of IgA1 that remained constant over time. This mirrors what we observed for anti-spike IgA titers (Fig 2), suggesting that the low amount of IgA present in the saliva was of IgA1 isotype that is more susceptible to protease cleavage by the local microbiota. Moreover, in the saliva of vaccinated subjects, we measured very low levels of anti-spike IgA2 isotype, even in SARS-CoV-2-Exp individuals, suggesting that vaccination does not restimulate a mucosal immunoglobulin response. Finally, in COVID-19 individuals, we found only anti-spike IgA1 in plasma and both anti-spike IgA1 and IgA2 (likely of mucosal origin) in saliva (Fig 4B), suggesting that only COVID-19 but not vaccination is able to induce a significant mucosal response.

In order to evaluate for how long antibodies would be found in the saliva, we analyzed SARS-CoV-2-specific IgG and IgA levels both in plasma and saliva at three months from vaccination (T3) in a limited cohort of naïve and SARS-CoV-2-exposed vaccinated subjects. Both anti-spike and anti-RBD IgG and IgA and anti-spike IgA1 in plasma diminished at T3 in all subjects, to a greater extent in naïve vaccinated individuals (Fig EV3 and Appendix Fig S1). This was paralleled by a reduction in the abundance of antibodies in saliva at three months from vaccination (Fig EV4 and Appendix Fig S2). As expected, we did not find SARS-COV-2-specific IgA2 in the plasma and only very low levels in the saliva (Figs EV3 and EV4; Appendix Figs S1 and S2).

In conclusion, BNT162b2 vaccine induces a strong immune response that permeates the saliva, and this may explain protection against viral infection at least after the second vaccine dose. However, this protection would not be attributed to IgA as they are probably not preserved in the saliva being of serum origin. The IgG may opsonize the virus and impede its infectivity indicating also why fully vaccinated individuals are hardly infective. We observed that IgG are clearly found in the saliva when they are present at high levels in the plasma and drop dramatically at three months from vaccination. This may explain why there is a waning of protection against infection in people vaccinated with the Pfizer vaccine at 4 months from vaccination without any drop on protection against hospitalization or death (Chemaitelly et al, 2021). We also identified a threshold of plasma SARS-CoV-2-specific IgG antibody over which permeation of SARS-CoV-2-specific IgG in the saliva occurs. We have not measured the presence of SARS-CoV-2-specific IgG and IgA in other mucosal fluids. However, two recent publications show that SARS-CoV-2-specific IgG and IgA are both detected also in nasal fluid samples (Chan et al, 2021; Guerrieri et al, 2021), even though at lower levels than in the saliva (Guerrieri et al, 2021). The fact that SARS-CoV-2-specific IgA2 were not detected in the saliva of

vaccinated subjects supports the need for mucosal vaccination and/or boosting strategies that promote anti-SARS-CoV-2 S-IgA, which are more potent at neutralizing the virus than IgGs thanks to their mechanism of pathogen clearance called immune exclusion (Mantis et al, 2011; Wang et al, 2021). This would be the most direct pathway to sterilizing immunity at the site of virus entry and replication.

# Materials and Methods

### Study approval

We tested the antibody response developed after the Pfizer/BioNTech vaccine (anti-SARS-CoV-2 IgGs, IgAs, IgA1, and IgA2 both in plasma and saliva) in 92 healthcare professionals of which 21 had a previous history of SARS-CoV-2 exposure (SARS-CoV-2-Exp) as part of two observational studies (Prot. Nr. CE Humanitas ex D.M. 8/2/2013 48/21 and CLI-PR-2102). Accrual was on a voluntary basis and health care and administrative staff were followed for serology after Pfizer/BioNTech vaccine at the time of the first vaccine dose (T0), at the time of the second dose (T1), 7–10 days from the second dose (T2), and three months after the second dose (T3). As a control, we tested the presence of anti-SARS-CoV-2 IgG, IgA, IgA1, and IgA2 antibodies (both in plasma and saliva) in 19 not vaccinated healthcare professionals ($n = 19$, Control) that were not exposed to SARS-CoV-2 (observational study ClinicalTrials.gov NCT04451577) and in a cohort of COVID-19 patients ($n = 28$, COVID-19) (ClinicalTrials.gov NCT04552340). All these studies were conducted at Istituto Clinico Humanitas and approved by the institutional review board of Istituto Clinico Humanitas. All participants signed an informed consent, and the experiments conformed to the principles set out in the WMA Declaration of Helsinki and the Department of Health and Human Services Belmont Report. For demographics characteristics, please refer to Appendix Table S2.

### Measurement of total IgA and IgG concentration within saliva samples

Commercial enzyme-linked immunosorbent assay (ELISA) kits were used to quantify the concentration of total IgG and IgA in saliva samples according to the manufacturer's instructions (Human IgA SimpleStep ELISA kit #ab196263 and Human IgG SimpleStep ELISA kit #ab195215, Abcam).

## Measurement of antigen-specific IgA, IgG, IgA1, and IgA2

The dual-ELISA protocol for sequential detection of anti-SARS-Cov-2 IgA and IgG antibodies, respectively, was adapted from a previously established single substrate ELISA protocol (Amanat *et al*, 2020). Wells of 96-well plates (Nunc Maxisorp; Thermo Fisher Scientific) were coated overnight at 4°C with 50 μl per well of a 2 μg/ml solution of each the following proteins suspended in PBS (Gibco): the SARS-Cov-2 full-length spike protein (cat#40589-V08B1, Sinobiological), receptor-binding domain (RBD) (cat#40592-V08H, Sinobiological), or nucleocapsid (N protein) (cat#40588-V08B, Sinobiological). The plates were then washed three times with 250 μl per well of 0.1% PBST using an automatic plate washer, and 100 μl per well of 3% non-fat milk prepared in PBS with 0.1% Tween 20 (PBST) was added to the plates at room temperature for 1 h as a blocking solution. Plasma and saliva samples were heated at 56°C for 30 min before use for virus inactivation. Plasma (1:1,000) and Saliva (1:10) samples were prepared in 1% non-fat milk in PBST. Sample dilutions for plasma and saliva were determined after set-up assays. The blocking solution was removed, and 50 μl of each serial dilution was added to the plates for 2 h at room temperature. Next, a mixture of two secondary antibodies was prepared in 0.1% PBST: Mouse anti-Human IgG-Fc horseradish peroxidase (HRP-) conjugated (1:5,000 dilution) (cat#A01854, Genescript) and Goat anti-Human IgA alkaline phosphatase (AP)-conjugated (1:16,000 dilution) (cat#A18790, Thermo Fisher Scientific). The secondary antibody mixture (50 μl) was added to each well for 1 h. Plates were washed again three times with 0.1% PBST.

To detect bound IgA in a first step, 50 μl of 1-Step™ PNPP Substrate Solution (Cat#37621, Thermo Fisher Scientific) was added to each well. To subsequently detect bound IgG, PNPP solution was discarded, and plates were washed once before addition of Ultra TMB-ELISA Substrate Solution (cat#34028, Thermo Fisher Scientific). The absorbance at 405 nm ($A_{405nm}$) and 450 nm ($A_{450nm}$) was measured using Eon™ (BioTek) plate reader.

The use of this sequential enzymatic reaction system (AP then HRP) did not affect the assay efficiency, which was identical to single enzyme reactions.

The same procedure was used to detect antigen-specific IgA1 and IgA2 isotypes in plasma and saliva. In this case, AP-conjugated mouse anti-Human IgA1 (Cat# 9130-04, Southernbiotech) and HRP-conjugated mouse anti-Human IgA2 (Cat# 9140-05, Southernbiotech) were used at 1:2,000 and 1:8,000 dilutions, respectively.

On each plate, two duplicates of a serial 1:2 dilution in PBS 1% milk of the World Health Organization (WHO) International Standard and Reference Panel for anti-SARS-CoV-2 antibody (NIBSC code: 20/136, UK) were used as standard curve, starting from a concentration of 30 IU/ml. Before quantifying antibody titers, we tested the performance of the anti-IgA and anti-IgG detection antibodies of WHO standard curves in saliva, as this is a complex matrix, and compared it to that of saline silution. We diluted the standard in three negative (T0) samples of saliva and compared it with a standard curve diluted in PBS 1% milk. The absorbance ($A_{405nm}$ or $A_{450nm}$) of the standard curve diluted in PBS 1% milk and in saliva were comparable: $r^2$ were consistently > 0.95 with no significant increase in background signal.

To calculate antibody titers, absorbance values of each experimental sample were interpolated with the average standard curve after correction for the absorption of blank (1% milk in PBS) controls.

The WHO International Standard is derived from COVID-19 patient's plasma; thus, it cannot be used as a standard reference for IgA2 measurement. Hence, we included a pool of saliva samples from COVID-19 patients in each plate to normalize the absorbance values ($A_{450nm}$) for IgA2 detection, and they are expressed in arbitrary unit (AU).

For saliva samples, the titers of antigen-specific Ig (IU/ml for IgA, IgA1, IgG and AU for IgA2) were normalized by dividing the values of SARS-CoV-2-specific Ig by total IgA or IgG concentrations of each sample. The normalization was applied only to values higher than LoD. The adjusted values are expressed in AU.

Limit of detection (LoD) of all our ELISA assays was calculated according to the procedure reported in the globally approved guideline CLSI EP17-A, published by Clinical Laboratory and Standards Institute (CLSI). Statistical analyses were performed on values higher than LoD (even though in the boxplots related to plasma samples, we showed all the values).

## Statistical analysis

Data were analyzed for normal distribution (Shapiro–Wilk test) before any statistical analyses. Statistical significance between different time points was determined using the Friedman test with the Dunn's multiple comparison test. Subjects with samples collected at all the time points were considered in the statistical analysis. The comparison of multiple groups was carried out using the Kruskal–Wallis test followed by the Dunn's multiple comparison test. A probability value of $P < 0.05$ was considered significant. All statistics are reported in the figure legends. Data analyses were carried out using GraphPad Prism version 8.

In order to gauge the linear relation between spike SARS-CoV-2-specific IgG values in plasma (y variable) and saliva (x variable), we estimate a linear regression model. Computations and visualization (Fig EV1A) are obtained by using Python programming language and, in particular, through the seaborn and SciPy libraries (Virtanen *et al*, 2020; Waskom, 2021). The plot in Fig EV1A also displays CI intervals at 95% obtained through bootstrap procedure (Bradley & Tibshirani, 1994). In order to estimate the threshold point, we computed the entire ROC curve obtaining combinations of specificity and sensitivity metrics for different cut-off values. To select the optimal threshold value, we computed the F1 score (Baeza-Yates & Ribeiro-Neto, 1999) and selected the value that maximized this metric. The F1 score, computed as a harmonic mean between specificity and sensitivity, is a popular metric in information retrieval and machine-learning literature that captures a discriminative threshold's ability to maximize sensitivity and specificity jointly. However, in its basic formulation, the F1 score gives equal importance to specificity and sensitivity. This property may be suboptimal in medical scenarios, but we carried out simple modifications (i.e., switching from a harmonic mean to a weighted harmonic mean) to account for different risk aversion levels.

**The paper explained**

**Problem**

BNT162b2 vaccine has been shown to protect both from COVID-19 disease and infection (94% after the second dose). How can this protection from infection be explained? As this vaccine is administered systemically, it is not expected to induce a mucosal response; however, the finding that also infection is strongly reduced in vaccinated people suggests that the immunoglobulins have reached mucosal surfaces for barrier protection.

**Results**

Here we show that BNT162b2 vaccine allows for the release SARS-CoV-2-specific Ig in the saliva. Saliva antibodies are likely deriving from the microvasculature of gingival crevices, and thus, they originate from the serum. IgG are more stable and steadily increase over time, whereas IgA (present in the plasma as protease-susceptible IgA1 isotype) are present at very low levels. Moreover, we show that SARS-CoV-2-specific Ig both in the saliva and in the plasma are almost completely lost at three months.

**Impact**

The undetectable mucosal response in the saliva of vaccinated subjects supports the need for mucosal vaccination and/or boosting strategies. The drop of IgG at three months from vaccination may explain why there is a waning of protection against infection in people vaccinated with the BNT162b2 vaccine at 4 months from vaccination.

## Data availability

This study includes no data deposited in external repositories.

**Expanded View** for this article is available online.

## Acknowledgements

This work was supported by the Italian Ministry of Health (Ricerca Finalizzata: COVID-2020-12371640 and Ricerca Corrente). We would like to thank all the employees and the patients who volunteered to participate to this study, all the vaccinating doctors, the nurses, and personnel who collected the samples and Humanitas Operations Management and Customer Care that coordinated vaccinations and blood draws. Synopsis figure was created with BioRender.com.

## Author contributions

**Maria Rescigno:** Conceptualization; Supervision; Funding acquisition; Writing—original draft; Writing—review and editing. **Abbass Darwich:** Data curation; Formal analysis; Methodology. **Chiara Pozzi:** Data curation; Formal analysis; Project administration; Writing—review and editing. **Giulia Fornasa:** Data curation; Formal analysis; Methodology; Writing—review and editing. **Michela Lizier:** Data curation; Formal analysis; Methodology; Writing—review and editing. **Elena Azzolini:** Conceptualization; Project administration. **Ilaria Spadoni:** Formal analysis; Methodology. **Francesco Carli:** Formal analysis; Methodology. **Antonio Voza:** Data curation; Formal analysis. **Antonio Desai:** Data curation; Formal analysis. **Carlo Ferrero:** Data curation; Methodology. **Luca Germagnoli:** Methodology. **Alberto Mantovani:** Conceptualization; Funding acquisition.

In addition to the CRediT author contributions listed above, the contributions in detail are:

AD performed the set-up and involved in ELISA analysis; CP contributed to data analysis and wrote the manuscript; EA coordinated the recruitment and sampling of subjects (project administration) and participated in clinical study design; GF contributed to data analysis, to ELISA assay and manuscript revision; ML and IS contributed to ELISA assay; FC contributed to LR and involved in F1 score analyses; AV and ADe involved in recruitment and data analysis of COVID-19 patients; CF and LG contributed to data analysis to calculate LoD; ICH COVID-19 task force: processing of plasma and saliva samples; AM involved in conceptualization and funding acquisition; MR conceived the study, analyzed the data, and wrote the manuscript.

## Disclosure and competing interests statement

Maria Rescigno is an EMBO Member and EMBO Council Member. This has no bearing on the editorial consideration of this article for publication. All other authors have no conflict of interest to declare.

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

## Appendix

ICH COVID-19 task-force: Aghemo Alessio, Anfray Clement, Badala-menti Salvatore, Belgiovine Cristina, Bertocchi Alice, Bombace Sara, Brescia Paola, Calcaterra Francesca, Calvi Michela, Cancellara Assunta, Capucetti Arianna, Carenza Claudia, Carloni Sara, Carne-vale Silvia, Cazzetta Valentina, Cecconi Maurizio, Ciccarelli Michele, Coianiz Nicolò, Darwich Abbass, Ana Lleo De Nalda, De Paoli Federica, Di Donato Rachele, Digifico Elisabeth, Durante Barbara, Farina Floriana Maria, Ferrari Valentina, Fornasa Giulia, Franzese Sara, Gil Gomez Antonio, Giugliano Silvia, Ana Rita Gomes, Lizier Michela, Lo Cascio Antonino, Melacarne Alessia, Mozzarelli Alessandro, My Ilaria, Oresta Bianca, Pasqualini Fabio, Pastò Anna, Pelamatti Erica, Perucchini Chiara, Pozzi Chiara, Rimoldi Valeria, Rimoldi Monica, Scarpa Alice, Selmi Carlo, Silvestri Alessandra, Sironi Marina, Spadoni Ilaria, Spano' Salvatore, Spata Gianmarco, Supino Domenico, Tentorio Paolo, Ummarino Aldo, Valentino Sonia, Voza Antonio, Zaghi Elisa, Zanon Veronica

