## [Review Process File · EMBO Molecular Medicine]

BNT162b2 vaccine induces antibody release in saliva: a possible role for mucosal viral protection?

Maria Rescigno, Abbass Darwich, Chiara Pozzi, Giulia Fornasa, Michela Lizier, Elena Azzolini, Ilaria Spadoni, Francesco Carli, Antonio Voza, Antonio Desai, Carlo Ferrero, Luca Germagnoli, and Alberto Mantovani

DOI: [10.15252/emmm.202115326](https://doi.org/10.15252/emmm.202115326)

Corresponding author: Maria Rescigno (maria.rescigno@hunimed.eu)

Review Timeline:

Submission Date:	22nd Oct 21
Editorial Decision:	11th Nov 21
Revision Received:	1st Mar 22
Editorial Decision:	23rd Mar 22
Revision Received:	29th Mar 22
Accepted:	1st Apr 22

Editor: Lise Roth

Transaction Report:

11th Nov 2021

Dear Prof. Rescigno,

Thank you for submitting your work to EMBO Molecular Medicine. We have now heard back from the referees who agreed to evaluate your manuscript. As you will see below, while reviewer #1 is overall supportive of publication of your study, reviewers #2 and #3 raise several issues, which unfortunately preclude its publication in EMBO Molecular Medicine in its current form.

In particular, referee #3 raised a potential limitation regarding novelty of the findings due to a recent publication (<https://doi.org/10.3389/fimmu.2021.744887>).

The referees were invited to cross-comment on the manuscript considering this publication, and after discussion within the editorial team, we agree that your findings have value in that they confirm and complement the previous findings, and we would therefore welcome the submission of a revised version of your manuscript with the understanding that all concerns from the referees should be addressed.

On this point, referee #2 stated:

"I was not aware of the Frontiers paper, I would tend now to ask them not only to discuss limitations, but simply to solve issues. I now stand by all the technical issues raised by ref3 that I disregarded for the sake of novelty."

If you feel you can satisfactorily address the points listed by the referees, you may wish to submit a revised version of your manuscript. Please attach a covering letter giving details of the way in which you have handled each of the referees' concerns. A revised manuscript will once again be subject to review, and we cannot guarantee at this stage that the eventual outcome will be favorable.

Revised manuscripts should be submitted within three months of a request for revision; they will otherwise be treated as new submissions, except under exceptional circumstances in which a short extension is obtained from the editor.

Should you find that the requested revisions are not feasible within the constraints outlined here and prefer, therefore, to submit your paper elsewhere, we would welcome a message to this effect.

We require:

- 1) A .docx formatted version of the manuscript text (including legends for main figures, EV figures and tables). Please make sure that the changes are highlighted to be clearly visible.
- 2) Individual production quality figure files as .eps, .tif, .jpg (one file per figure). For guidance, download the 'Figure Guide PDF' (<https://www.embopress.org/page/journal/17574684/authorguide#figureformat>).
- 3) A .docx formatted letter INCLUDING the reviewers' reports and your detailed point-by-point responses to their comments. As part of the EMBO Press transparent editorial process, the point-by-point response is part of the Review Process File (RPF), which will be published alongside your paper.
- 4) A complete author checklist, which you can download from our author guidelines (<https://www.embopress.org/page/journal/17574684/authorguide#submissionofrevisions>). Please insert information in the checklist that is also reflected in the manuscript. The completed author checklist will also be part of the RPF.
- 5) It is mandatory to include a 'Data Availability' section after the Materials and Methods. Before submitting your revision, primary datasets produced in this study need to be deposited in an appropriate public database, and the accession numbers and database listed under 'Data Availability'. Please remember to provide a reviewer password if the datasets are not yet public (see <https://www.embopress.org/page/journal/17574684/authorguide#dataavailability>).

- 6) For data quantification: please specify the name of the statistical test used to generate error bars and P values, the number (n) of independent experiments (specify technical or biological replicates) underlying each data point and the test used to calculate p-values in each figure legend. The figure legends should contain a basic description of n, P and the test applied. Graphs must include a description of the bars and the error bars (s.d., s.e.m.).

7) We would also encourage you to include the source data for figure panels that show essential data. Numerical data should be provided as individual .xls or .csv files (including a tab describing the data). For blots or microscopy, uncropped images should be submitted (using a zip archive if multiple images need to be supplied for one panel). Additional information on source data and instruction on how to label the files are available at .

8) Our journal encourages inclusion of *data citations in the reference list* to directly cite datasets that were re-used and obtained from public databases. Data citations in the article text are distinct from normal bibliographical citations and should directly link to the database records from which the data can be accessed. In the main text, data citations are formatted as follows: "Data ref: Smith et al, 2001" or "Data ref: NCBI Sequence Read Archive PRJNA342805, 2017". In the Reference list, data citations must be labeled with "[DATASET]". A data reference must provide the database name, accession number/identifiers and a resolvable link to the landing page from which the data can be accessed at the end of the reference. Further instructions are available at .

9) We replaced Supplementary Information with Expanded View (EV) Figures and Tables that are collapsible/expandable online. A maximum of 5 EV Figures can be typeset. EV Figures should be cited as 'Figure EV1, Figure EV2' etc... in the text and their respective legends should be included in the main text after the legends of regular figures.

10) The paper explained: EMBO Molecular Medicine articles are accompanied by a summary of the articles to emphasize the major findings in the paper and their medical implications for the non-specialist reader. Please provide a draft summary of your article highlighting

11) For more information: There is space at the end of each article to list relevant web links for further consultation by our readers. Could you identify some relevant ones and provide such information as well? Some examples are patient associations, relevant databases, OMIM/proteins/genes links, author's websites, etc...

12) Every published paper now includes a 'Synopsis' to further enhance discoverability. Synopses are displayed on the journal webpage and are freely accessible to all readers. They include a short stand first (maximum of 300 characters, including space) as well as 2-5 one-sentences bullet points that summarizes the paper. Please write the bullet points to summarize the key NEW findings. They should be designed to be complementary to the abstract - i.e. not repeat the same text. We encourage inclusion of key acronyms and quantitative information (maximum of 30 words / bullet point). Please use the passive voice. Please attach these in a separate file or send them by email, we will incorporate them accordingly.

13) As part of the EMBO Publications transparent editorial process initiative (see our Editorial at <http://embomolmed.embopress.org/content/2/9/329>), EMBO Molecular Medicine will publish online a Review Process File (RPF) to accompany accepted manuscripts.

In the event of acceptance, this file will be published in conjunction with your paper and will include the anonymous referee reports, your point-by-point response and all pertinent correspondence relating to the manuscript. Let us know whether you agree with the publication of the RPF and as here, if you want to remove or not any figures from it prior to publication. Please note that the Authors checklist will be published at the end of the RPF.

I look forward to seeing a revised form of your manuscript as soon as possible. Use this link to login to the manuscript system

and submit your revision: Link Not Available

Yours sincerely,

Lise Roth

Lise Roth, PhD
Editor
EMBO Molecular Medicine

***** Reviewer's comments *****

Referee #1 (Remarks for Author):

In this excellent paper, Rescigno and colleagues show that upon vaccination IgG are found also in the saliva, and more abundant in SARS-CoV-2 previously exposed subjects, paralleling the development of serum IgG. These antibodies fade at 3 months from vaccination. They identified a threshold of plasma IgG over which they permeate in the saliva. Regarding IgA antibodies, we found only protease-susceptible IgA1 antibodies in plasma while they were not present in the saliva over the course of vaccination of SARS-CoV-2 naïve subjects. Thus, in response to BNT162b2 vaccine, serum IgG can permeate into mucosal sites and participate to viral protection.

This is a very well performed and very timely study that will be of great interest to a very wide readership. The paper should be published with our delay. A few very minor suggestions:

1. In all the figures the Ab response is shown as OD values. Could the authors present the data as Reciprocal EC50 titers?
2. Figure 1: On the x-axis, please indicate the time points at which the serum and saliva Ab titers were measured at. The figure nor the legend provides no such information. The same applies to all the other figures.
3. Figure 2: On the x-axis, please indicate the time points at which the serum and saliva Ab titers were measured at. The figure nor the legend provides no such information. The same applies to all the other figures.
4. Figure 3: The vaccinated subjects were assayed 7 to 10 days after the 2nd dose. However other studies have shown that the antibody titers to BNT162b2 vaccination continue to increase up to 21 days following secondary vaccination and beyond. Can the authors provide any data at day 42 or some such time point?

Referee #2 (Remarks for Author):

Useful informations are reported in this paper since available data on mucosal isotypes induced by vaccination is scarce. Although the authors face obvious issues of high background and low sensitivity of their saliva IgA assay, it is quite clear that IgA2 are much less present than IgA1 in saliva following, and that both of them are difficult to detect at late times. In contrast IgA2 appear induced (probably locally) by infection, as expected. The argument that IgA antibody is released in saliva and not induced is plausible, although not directly demonstrated.

Comments:

- 1/ I have a problem with the title. It cannot be stated that "BNT162b2 vaccine induces antibody release in saliva for mucosal viral protection" since protection is neither studied in that paper. We do not know whether vaccinees with high saliva IgA levels are better protected. I agree that it is expected, but not demonstrated. Protection should be removed from the title.
- 2/ The data presented are not quantitative (only optical densities), it should be underlined as a limitation
- 3/ The data presented are limited to saliva, while nasal and lung are the big issue, and we know that the balance between mucosal isotypes is different there, it should be underlined as a limitation
- 4/ the authors need to show that their dual assay (detection of IgA, then IgG in same well) is not problematic. It could be that IgA compete with IgG?

Referee #3 (Remarks for Author):

This study seems to present some potentially interesting results in the area of antibody "levels" in saliva. This reviewer notes a recent publication which has some parallels with this one but is not mentioned or discussed.

(<https://doi.org/10.3389/fimmu.2021.744887>

). However, several issues need to be addressed in order to rely upon the authors' conclusions. As it stands this is more of a preliminary report with several reservations.

No ethical issues to note.

MAJOR ISSUES:

1) Relatively small numbers of samples analysed.

2) Authors mention high backgrounds which could undermine interpretation.

3) No quantification of amount of protein assayed in saliva samples from different individuals (which can vary in terms of overall protein concentration quite widely- simply stating a volume does not provide any sort of cross-sample standardisation e.g. in terms of ug of total protein in each sample or some similar standardisation process). See also many papers on variation of protein conc in saliva e.g. <https://doi.org/10.1177/10454411950060040501>
<https://doi.org/10.3389/fphys.2019.00652>

For example "the amount of protein extracted from saliva covers a wide e.g. ranged from 0.07 to 0.95 mg protein per mL saliva <https://doi.org/10.1007/s12263-009-0121-x>

It seems to me that these considerations undermine any conclusions drawn about specific IgA fluctuations. It should be possible to measure total IgA and or total IgG in each saliva sample and then normalise the cognate IgA/IgG response to total.

4) Relatively short follow up times after second dose (7- 10 days)- seems quite short in the context of following development of boosting an IgG/IgA response.

5) Some of the conclusions and data seem rather confirmatory rather than ground-breaking in nature.

6) Some conclusions are made upon data which is acknowledged to have no statistical significance e.g. line 141-142

Less important issues that nevertheless should be considered

7) Lack of detection limit controls e.g. Line 146 authors state " This may justify why IgG are detectable in the saliva while IgA are not". What are the detection limits of your assays? The presence of data showing what the detection limit is for ANY IgA in the samples would be useful. A further complication is the documented sensitivity of IgA1 to bacterial (Streptococcal, Haemophilus and Neisseria) - could presence of these organisms interfere with the IgA1 assays being run?

8) A general point, many parts of the manuscript refer to unquantified changes e.g. line 152 "a strong reduction in the abundance of antibodies in saliva at three months" - I think it is good practice to give the magnitudes of these changes e.g. 80% reduction or 3 fold reduction or whatever that than use extremely ambiguous and non-quantifiable descriptions such as "strong".

Specific issues of varying significance (some simply illustrative of points 1-8 above).

Line 34 replace "is protecting also" with "also protects"

Line 35 replace "IgG are found also in " with "IgG is also found in " or possibly more precise language such as "cognate immunoglobulin G molecules are also found in"

Line 37 "These antibodies fade" is rather unscientific- I assume authors mean the antibody titre declines?

Line 37-38. The statement "We identified a threshold of plasma IgG over which they permeate in the saliva" Is rather vague. Do you refer to a concentration of specific IgG above which the relevant IgG can be detected in the saliva? Please clarify.

Line 39 "proetase" - should be proteinase or protease.

Line 41 "participate to viral protection " should be " participate in viral protection "

Line 41 What is the basis for this conclusion? "IgA1 are likely degraded in the saliva. " Bacteria such as Neisseria, Streptococcus and Haemophilus are known to produce such proteases, but are the authors suggesting instability of IgA1 is a particular issue in anti-Spike IgA1 -this is confusing or might some samples be infected with one or other of these organisms? Carriage rates can be high depending on the season.

Line 49 hyphenate "mRNA-based"

Line 56 The authors discuss the efficacy of the Pfizer/BioNTech vaccine - they seem to ignore that the vaccine has lower efficacy against more recent forms of the virus such as Delta. This should at least be mentioned in passing.

Line 83 The authors show that "spike and RBD steadily increased in the plasma and saliva of SARS-CoV-2 naïve subjects after first an second vaccine dose" This is hardly surprising is it? Similarly, the authors state that other findings (line 86) are "in agreement with what we and others previously published on plasma IgG". This suggests that at least some of the work presented is rather confirmatory in nature.

Lines 94 - 96 suggest that this study may be somewhat flawed, in that the authors admit to not knowing whether the high level of anti-spike IgA in the saliva (before vaccination) was "background noise"

Line 100. The authors make the statement "Interestingly, we found a correlation between the amount of IgG in plasma versus

that in the saliva." - Is this anything out of the ordinary? One would imagine that this is quite the normal relationship between plasma IgG conc and that in the saliva or is this a surprising finding? Again, one assumes the authors mean immunoglobulins specific for the antigen in question, but this is not explicitly stated.

103- Maybe the "threshold" they refer to is due to the poor signal to noise they mention in lines 94-96?

Line 129- Authors make a rather broad claim "These data strongly support the notion that antibodies from the plasma permeate mucosal fluids, particularly the saliva where serum antibodies can filter via the microvasculature of gingival crevices." As the authors tested only one type of mucosal fluid- saliva, they might consider reducing the scope of this claim. They didn't look in other mucosal fluids?

Line 161 " or death (Chemaitelly et al, 2021).We also" missing space after full stop.

Line 161 "We also identified a threshold of antibody titers over which permeation of the IgG in the saliva occurs." Perhaps you could state what that threshold was?

Line 164 "anti-SARS-CoV-2 S-IgA, which are more potent at neutralizing the virus than IgGs" Perhaps the authors might point out that IgA operates via the immune exclusion route and add in an appropriate reference?

355 Fig 3 and elsewhere. Please spell "length" correctly, spelling is incorrect in some figures.

Supplementary Fig 3 legend makes no mention of panel A or panel B, yet two panels (A and B) appear in the figure? Please correct. Spelling incorrect (Full-lenght Spike) - length ! Probably best to check the entire manuscript for this mistake.

Referee #1 (Remarks for Author):

In this excellent paper, Rescigno and colleagues show that upon vaccination IgG are found also in the saliva, and more abundant in SARS-CoV-2 previously exposed subjects, paralleling the development of serum IgG. These antibodies fade at 3 months from vaccination. They identified a threshold of plasma IgG over which they permeate in the saliva. Regarding IgA antibodies, we found only protease-susceptible IgA1 antibodies in plasma while they were not present in the saliva over the course of vaccination of SARS-CoV-2 naïve subjects. Thus, in response to BNT162b2 vaccine, serum IgG can permeate into mucosal sites and participate to viral protection.

This is a very well performed and very timely study that will be of great interest to a very wide readership. The paper should be published with our delay. A few very minor suggestions:

1. In all the figures the Ab response is shown as OD values. Could the authors present the data as Reciprocal EC50 titers?

To address this critical issue raised by the reviewers, we repeated all of the ELISA assays including the standard control in order to quantify the Ab response (now shown in IU/ml for plasma samples).

We used the World Health Organization (WHO) International Standard and Reference Panel for anti-SARS-CoV-2 antibody (NIBSC, UK) as standard curve for the quantification of the antibody titers in plasma and saliva. Before doing so however, we tested the performance of the anti-IgA and anti-IgG detection antibodies of WHO standard curves in saliva, as this is a complex matrix, and compared it to that of saline dilution. We diluted the standard in three negative (T0) samples of saliva, and compared with a standard curve diluted in PBS with 1% milk. The optical densities (OD) of the standard curve diluted in PBS 1% milk and in saliva were comparable: r^2 were consistently > 0.95 with no significant increase in background signal.

Figure for reviewers removed

2. Figure 1: On the x-axis, please indicate the time points at which the serum and saliva Ab titers were measured at. The figure nor the legend provides no such information. The same applies to all the other figures.

Thank you for the suggestion. We have added this information in the figure legends.

3. Figure 2: On the x-axis, please indicate the time points at which the serum and saliva Ab titers were measured at. The figure nor the legend provides no such information. The same applies to all the other figures.

Thank you for the suggestion. We have added this information in the figure legends.

4. Figure 3: The vaccinated subjects were assayed 7 to 10 days after the 2nd dose. However other studies have shown that the antibody titers to BNT162b2 vaccination continue to increase up to 21 days following

secondary vaccination and beyond. Can the authors provide any data at day 42 or some such time point?

We collected plasma and saliva only 7-10 days after the 2nd dose but the induction of Ab response was already clear. Moreover, we measured the Ab response in the plasma and saliva 3 months after the 2nd dose.

Referee #2 (Remarks for Author):

Useful informations are reported in this paper since available data on mucosal isotypes induced by vaccination is scarce. Although the authors face obvious issues of high background and low sensitivity of their saliva IgA assay, it is quite clear that IgA2 are much less present than IgA1 in saliva following, and that both of them are difficult to detect at late times. In contrast IgA2 appear induced (probably locally) by infection, as expected. The argument that IgA antibody is released in saliva and not induced is plausible, although not directly demonstrated.

Comments:

1/ I have a problem with the title. It cannot be stated that "BNT162b2 vaccine induces antibody release in saliva for mucosal viral protection" since protection is neither studied in that paper. We do not know whether vaccinees with high saliva IgA levels are better protected. I agree that it is expected, but not demonstrated. Protection should be removed from the title.

Thank you for the suggestion. We changed the title: "BNT162b2 vaccine induces antibody release in saliva: a possible role for mucosal viral protection?".

2/ The data presented are not quantitative (only optical densities), it should be underlined as a limitation

To address this critical issue raised by the reviewers, we repeated all of the ELISA assays including the standard control in order to quantify the Ab response (now shown in IU/ml for plasma samples).

We used the World Health Organization (WHO) International Standard and Reference Panel for anti-SARS-CoV-2 antibody (NIBSC, UK) as standard curve for the quantification of the antibody titers in plasma and saliva. Before doing so however, we tested the performance of the anti-IgA and anti-IgG detection antibodies of WHO standard curves in saliva, as this is a complex matrix, and compared it to that of saline dilution. We diluted the standard in three negative (T0) samples of saliva, and compared it with a standard curve diluted in PBS plus 1% milk. The optical densities (OD) of the standard curve diluted in PBS 1% milk and in saliva were comparable: r^2 were consistently > 0.95 with no significant increase in background signal.

Figure for reviewers removed

3/ The data presented are limited to saliva, while nasal and lung are the big issue, and we know that the balance between mucosal isotypes is different there, it should be underlined as a limitation

Thank you for the suggestion. We have added this point as a limitation in the discussion. (Lines 168-172).

4/ the authors need to show that their dual assay (detection of IgA, then IgG in same well) is not problematic. It could be that IgA compete with IgG?

We thank the reviewer for this important inquiry. Indeed, when setting up the dual assay we had to verify two main points:

1. Regarding the compatibility of the two enzymatic reactions (AP + PNPP and then HRP + TMB): To answer this, we coated plates directly with secondary antibodies conjugated with AP enzyme or HRP

enzyme alone or together and we compared the single read with the sequential read. As can be seen below, reading HRP reaction after AP reaction does not cause any loss or gain of unspecific signal.

- Regarding a possible interference between IgA and IgG (Steric hindrance or other): To answer this, we compared the two formats of the assay on a set of plasma samples from healthy and COVID-19 positive individuals (with very low, medium, and high antibody titers according to DiaSorin's LIAISON test¹). The data below shows that the results from the dual assay are undistinguishable from those of single measurement IgA and IgG.

- Altawalah, H., Alfouzan, W., Al-Fadalah, T. & Ezzikouri, S. Diagnostic performance of automated SARS-CoV-2 antigen assay in nasal swab during COVID-19 vaccination campaign. *Diagnostics* (2021) doi:10.3390/diagnostics11112110.

Referee #3 (Remarks for Author):

This study seems to present some potentially interesting results in the area of antibody "levels" in saliva. This reviewer notes a recent publication which has some parallels with this one but is not mentioned or discussed. (<https://doi.org/10.3389/fimmu.2021.744887>). However, several issues need to be addressed in order to rely upon the authors' conclusions. As it stands this is more of a preliminary report with several reservations.

We have now mentioned this and other more recent publications.

No ethical issues to note.

MAJOR ISSUES:

1) Relatively small numbers of samples analysed.

We have increased the number of the samples analyzed.

In particular:

- we analyzed antigen specific IgA1 and IgA2 in the plasma and saliva of about 85 vaccinated individuals (instead of 25 as in the previous analysis)
- we measured the Ab response (not only IgG, IgA, but also IgA1, IgA2) at 3 months from second dose (T3) in the plasma of 56 vaccinated subjects (instead of 24 as in the previous analysis) and in the saliva of 76 vaccinated individuals (instead of 50 as in the previous analysis).
- we analyzed SARS-CoV-2 specific IgG and IgA in the plasma and saliva of 92 and 85 vaccinated subjects, respectively (instead of 83 as in the previous analysis).
- as we had to repeat all of the ELISA assays including the standard control in order to quantify the Ab response, some samples from COVID19 patients were finished. So, we measured antigen specific IgG, IgA, IgA1 and IgA2 in the plasma and saliva of 28 COVID-19 patients.

2) Authors mention high backgrounds which could undermine interpretation.

To solve this problem, we calculated the limit of detection (LoD) of all of our 16 different ELISA assays according to the procedure reported in the globally approved guideline CLSI EP17-A, published by Clinical Laboratory and Standards Institute (CLSI) and we performed statistical analysis only on values higher than LoD. See Methods section.

3) No quantification of amount of protein assayed in saliva samples from different individuals (which can vary in terms of overall protein concentration quite widely- simply stating a volume does not provide any sort of cross-sample standardisation e.g. in terms of ug of total protein in each sample or some similar standardisation process). See also many papers on variation of protein conc in saliva

e.g. <https://doi.org/10.1177/10454411950060040501>

<https://doi.org/10.3389/fphys.2019.00652>

For example "the amount of protein extracted from saliva covers a wide e.g. ranged from 0.07 to 0.95 mg protein per mL saliva <https://doi.org/10.1007/s12263-009-0121-x>

It seems to me that these considerations undermine any conclusions drawn about specific IgA fluctuations. It should be possible to measure total IgA and or total IgG in each saliva sample and then normalise the cognate IgA/IgG response to total.

Thank you for the suggestion. We measured total IgA and total IgG in each saliva sample using commercial enzyme-linked immunosorbent assays (ELISA). Then, the titers of SARS-CoV-2 antigen specific Ig (IgG, IgA, IgA1, IgA2) were normalized by dividing their values by total IgA or IgG concentrations of each sample. The adjusted values are expressed in AU. See Methods section.

Moreover, we repeated all of the ELISA assays including the standard control in order to quantify the Ab response (now shown in IU/ml for plasma samples). We used the World Health Organization (WHO) International Standard and Reference Panel for anti-SARS-CoV-2 antibody (NIBSC, UK) as standard curve for the quantification of the antibody titers in plasma and saliva. Before doing so however, we tested the performance of the anti-IgA and anti-IgG detection antibodies of WHO standard curves in saliva, as this is a complex matrix, and compared it to that of saline dilution. We diluted the standard in three negative (T0) samples of saliva, and compared with a standard curve diluted in PBS with 1% milk. The optical densities (OD) of the standard curve diluted in PBS 1% milk and in saliva were comparable: r^2 were consistently > 0.95 with no significant increase in background signal.

Figure for reviewers removed

4) Relatively short follow up times after second dose (7- 10 days)- seems quite short in the context of following development of boosting an IgG/IgA response.

The reviewer is right, but unfortunately, we collected plasma and saliva only 7-10 days after the 2nd dose. However the induction of Ab response was already clear. Moreover, we measured the Ab response in the plasma and saliva 3 months after the 2nd.

5) Some of the conclusions and data seem rather confirmatory rather than ground-breaking in nature.

We think that the finding that IgA in the plasma are of the IgA1 isotype confirms that they have a systemic origin. This may explain why we can detect only IgG in the saliva as IgA1 are protease sensitive and thus we expect them to be unstable in the presence of bacterial proteases. We also followed the level of immunoglobulins after a long period of time and show that they drop in the saliva and this may explain why vaccinated people are protected from infection only early after vaccination. This has not been studied in the other publications and we think helps explaining why the vaccine protects from infection only in the first three months. We also identified a threshold of SARS-CoV-2 antigen specific IgG over which cognate IgG can be found in the saliva, this is also novel and suggests that individuals with amounts of SARS-CoV-2 antigen specific plasma IgG lower than the threshold may not be protected from infection.

6) Some conclusions are made upon data which is acknowledged to have no statistical significance e.g. line 141-142

We have rephrased it.

Less important issues that nevertheless should be considered

7) Lack of detection limit controls e.g. Line 146 authors state " This may justify why IgG are detectable in the saliva while IgA are not". What are the detection limits of your assays? The presence of data showing what the detection limit is for ANY IgA in the samples would be useful. A further complication is the

documented sensitivity of IgA1 to bacterial (Streptococcal, Haemophilus and Neisseria) - could presence of these organisms interfere with the IgA1 assays being run?

We have now calculated the limit of detection (LoD) of all of our 16 different ELISA assays according to the procedure reported in the globally approved guideline CLSI EP17-A, published by Clinical Laboratory and Standards Institute (CLSI) and we performed statistical analysis only on values higher than LoD. IgA and IgA1 levels showed a similar trend both in the plasma and in the saliva of vaccinated subjects suggesting that bacteria do not interfere with the IgA1 assays. This result suggests that all of the IgA present are IgA1. On the contrary, we did not find detectable levels of IgA2 in the plasma (as expected) of any individual, but only in the saliva of very few vaccinated individuals and COVID-19 patients.

8) A general point, many parts of the manuscript refer to unquantified changes e.g. line 152 "a strong reduction in the abundance of antibodies in saliva at three months" - I think it is good practice to give the magnitudes of these changes e.g. 80% reduction or 3 fold reduction or whatever that than use extremely ambiguous and non-quantifiable descriptions such as "strong".

Thank you for the suggestion. We have corrected this point.

Specific issues of varying significance (some simply illustrative of points 1-8 above).

Line 34 replace "is protecting also" with "also protects"

Thank you for the suggestion. We have corrected it.

Line 35 replace "IgG are found also in " with "IgG is also found in " or possibly more precise language such as "cognate immunoglobulin G molecules are also found in"

We have corrected it.

Line 37 "These antibodies fade" is rather unscientific- I assume authors mean the antibody titre declines?

We have corrected it.

Line 37-38. The statement "We identified a threshold of plasma IgG over which they permeate in the saliva" Is rather vague. Do you refer to a concentration of specific IgG above which the relevant IgG can be detected in the saliva? Please clarify.

We rephrased it.

Line 39 "proetase" - should be proteinase or protease.

We have corrected it.

Line 41 "participate to viral protection " should be " participate in viral protection "

Thank you. We have corrected it.

Line 41 What is the basis for this conclusion? "IgA1 are likely degraded in the saliva. " Bacteria such as Neisseria, Streptococcus and Haemophilus are known to produce such proteases, but are the authors suggesting instability of IgA1 is a particular issue in anti-Spike IgA1 -this is confusing or might some samples be infected with one or other of these organisms? Carriage rates can be high depending on the season.

We meant that since the saliva is full of microbes of the local microbiota and many of them release proteases capable of degrading IgA1 I am not surprised that they may be unstable once they are released from the gingival crevicular fluid (from the serum) and reach the saliva. We have now explained better what we meant.

Line 49 hyphenate "mRNA-based"

We have corrected it.

Line 56 The authors discuss the efficacy of the Pfizer/BioNTech vaccine - they seem to ignore that the vaccine has lower efficacy against more recent forms of the virus such as Delta. This should at least be mentioned in passing.

We have commented this point and added 2 references in the manuscript (Lines 57-60).

Line 83 The authors show that "spike and RBD steadily increased in the plasma and saliva of SARS-CoV-2 naïve subjects after first and second vaccine dose" This is hardly surprising is it? Similarly, the authors state that other findings (line 86) are "in agreement with what we and others previously published on plasma IgG". This suggests that at least some of the work presented is rather confirmatory in nature.

Nobody has actually analyzed in one single paper IgG, IgA1 and IgA2 in both plasma and saliva, thus it was not obvious that IgG followed a similar path, indeed IgA do not. Our finding that IgA in the plasma are of the IgA1 isotype tells us that they are of systemic origin and thus one could infer that there is no activation of a mucosal immune response. This may also explain why IgA are found at very low levels in the saliva of naïve subjects as being more protease sensitive they may be cleaved by proteases of the microbiota. To highlight that this is not confirmatory for all of the findings, we have thus modified the phrase and added: "However this was not true for the SARS-CoV-2 specific IgA response" (Lines 98-99).

Lines 94 - 96 suggest that this study may be somewhat flawed, in that the authors admit to not knowing whether the high level of anti-spike IgA in the saliva (before vaccination) was "background noise" .

We eliminated this sentence. Indeed, as commented above, to solve this problem, we calculated the limit of detection (LoD) of all our ELISA assays according to the procedure reported in the globally approved guideline CLSI EP17-A, published by Clinical Laboratory and Standards Institute (CLSI) and we performed statistical analysis only on values higher than LoD.

Line 100. The authors make the statement "Interestingly, we found a correlation between the amount of IgG in plasma versus that in the saliva." - Is this anything out of the ordinary? One would imagine that this is quite the normal relationship between plasma IgG conc and that in the saliva or is this a surprising finding? Again, one assumes the authors mean immunoglobulins specific for the antigen in question, but this is not explicitly stated.

The reviewer is right. We clarified this point (Lines 105-108).

103- Maybe the "threshold" they refer to is due to the poor signal to noise they mention in lines 94-96?

We clarified this point (Lines 105-108).

Line 129- Authors make a rather broad claim "These data strongly support the notion that antibodies from the plasma permeate mucosal fluids, particularly the saliva where serum antibodies can filter via the microvasculature of gingival crevices." As the authors tested only one type of mucosal fluid- saliva, they might consider reducing the scope of this claim. They didn't look in other mucosal fluids?

Unfortunately we collected only plasma and saliva samples and we did not look in other mucosal fluids. We have corrected the sentence by saying : These data strongly support the notion that antibodies from the plasma permeate ~~mucosal fluids, particularly~~ the saliva where serum antibodies can filter via the microvasculature of gingival crevices. (Line 131).

Line 161 " or death (Chemaitelly et al, 2021).We also" missing space after full stop.

We corrected it.

Line 161 "We also identified a threshold of antibody titers over which permeation of the IgG in the saliva occurs." Perhaps you could state what that threshold was?

We rephrased it (Line 167-168).

Line 164 "anti-SARS-CoV-2 S-IgA, which are more potent at neutralizing the virus than IgGs" Perhaps the authors might point out that IgA operates via the immune exclusion route and add in an appropriate reference?

We clarified this point and we added an appropriate reference (Line 175).

355 Fig 3 and elsewhere. Please spell "length" correctly, spelling is incorrect in some figures.

Supplementary Fig 3 legend makes no mention of panel A or panel B, yet two panels (A and B) appear in the figure? Please correct. Spelling incorrect (Full-lenght Spike) - length ! Probably best to check the entire manuscript for this mistake.

We checked and correct it.

23rd Mar 2022

Dear Prof. Rescigno,

Thank you for the submission of your manuscript to EMBO Molecular Medicine. We have now received the enclosed reports from the two referees who re-reviewed your manuscript. As you will see, they are supportive of publication pending minor revisions, and I am therefore pleased to inform you that we will be able to accept your manuscript once the following points will be addressed:

1/ Please address the remaining minor concerns raised by referees #2 and #3.

2/ Manuscript text file:

- Please accept the minor changes added by our data editors in the main manuscript file labelled 'Data edited MS file'. Please use this file for any further modification (in track changes mode).
- Please remove the yellow highlighted and underlined text.
- We can accommodate up to 5 keywords. Would you like to add SARS-CoV-2 and/or BNT162b2 and/or mucosal immunity (or other) to the keywords you currently have?
- Please remove "data not shown" (p. 9). As per our guidelines on "Unpublished Data", the journal does not permit citation of "Data not shown/unpublished data". All data referred to in the paper should be displayed in the main or Expanded View figures.
- Methods: In the Study approval section, please include the full statement that the experiments conformed to the principles set out in the WMA Declaration of Helsinki and the Department of Health and Human Services Belmont Report.
- Author contributions: please differentiate between Abbass Darwich and Antonio Desai (both A. D.).
- Disclosure statement and competing interests: Please rename this section. We updated our journal's competing interests policy in January 2022 and request authors to consider both actual and perceived competing interests. Please review the policy <https://www.embopress.org/competing-interests> and update your competing interests. In particular, please consider the following: "Editorial board members, EMBO council members, EMBO publications advisory board members, and EMBO staff must disclose their relationship with EMBO in the author disclosure statement using the standard phrase, "[Author] is an editorial advisory board/EMBO Member. This has no bearing on the editorial consideration of this article for publication."

3/ Please provide a synopsis text and figure. Synopses include a short stand first (maximum of 300 characters, including space) as well as 2-5 one-sentences bullet points that summarizes the paper. Please write the bullet points to summarize the key NEW findings. They should be designed to be complementary to the abstract - i.e. not repeat the same text. We encourage inclusion of key acronyms and quantitative information (maximum of 30 words / bullet point). Please use the passive voice.

Please also suggest a striking image or visual abstract to illustrate your article as a jpeg/tiff/png file 550 px wide x 300-600 px high.

4/ As part of the EMBO Publications transparent editorial process initiative (see our Editorial at <http://embomolmed.embopress.org/content/2/9/329>), EMBO Molecular Medicine will publish online a Review Process File (RPF) to accompany accepted manuscripts.

This file will be published in conjunction with your paper and will include the anonymous referee reports, your point-by-point response and all pertinent correspondence relating to the manuscript. Let us know whether you agree with the publication of the RPF and as here, if you want to remove or not any figures from it prior to publication.

I look forward to receiving your revised manuscript.

With kind regards,

Lise Roth

Lise Roth, PhD
Editor
EMBO Molecular Medicine

To submit your manuscript, please follow this link:

Link Not Available

The system will prompt you to fill in your funding and payment information. This will allow Wiley to send you a quote for the article processing charge (APC) in case of acceptance. This quote takes into account any reduction or fee waivers that you may be eligible for. Authors do not need to pay any fees before their manuscript is accepted and transferred to our publisher.

**** Reviewer's comments ****

Referee #2 (Comments on Novelty/Model System for Author):

There are already several papers looking at vaccine-induced saliva response, but the detection of all antibody isotypes, including IgA1 and A2 is novel
Technically the authors are obviously limited by the sensitivity for detection of IgA in saliva. At this point we do not know for sure whether there is an "absence of an IgA response in the saliva of naïve vaccinated subjects". The term absence might be too strong and this final statement should be mitigated

Referee #2 (Remarks for Author):

The authors nicely addressed the issues I raised.
Technically the authors are obviously limited by the sensitivity and background issues of their assay for detection of IgA in saliva. At this point we do not know for sure whether there is an "absence of an IgA response in the saliva of naïve vaccinated subjects". The term absence might be too strong and this final statement should be mitigated. I think a proper conclusion should be: not detectable using our technical setting, or something of the kind

Referee #3 (Comments on Novelty/Model System for Author):

The authors appear to have addressed all technical reservations expressed at first review and o have expanded the manuscript acceptably.

Referee #3 (Remarks for Author):

The authors appear to have addressed all technical reservations expressed at first review and the conclusions drawn seem well supported by their data.

Minor issues

Line 111. Authors report " 0.9374 +/- 0.02597" Can you really justify so many decimal places in the numbers reported? I am sure this is what the stats package reports but this is really 0.94 +/- 0.03 otherwise you are implying a precision of your measurements at 0.0001 which I doubt is the case.

Line 221 - Bad form to start sentences with Arabic numbers. Rephrase e.g. The secondary antibody mixture (50 μ L) ...

Throughout: I believe that OD is a defunct term and that most biochemical journals prefer absorbance e.g. A540nm (number/unit as subscript)?

***** Reviewer's comments *****

Referee #2 (Comments on Novelty/Model System for Author):

There are already several papers looking at vaccine-induced saliva response, but the detection of all antibody isotypes, including IgA1 and A2 is novel

Technically the authors are obviously limited by the sensitivity for detection of IgA in saliva. At this point we do not know for sure whether there is an "absence of an IgA response in the saliva of naïve vaccinated subjects". The term absence might be too strong and this final statement should be mitigated

Referee #2 (Remarks for Author):

The authors nicely addressed the issues I raised.

Technically the authors are obviously limited by the sensitivity and background issues of their assay for detection of IgA in saliva. At this point we do not know for sure whether there is an "absence of an IgA response in the saliva of naïve vaccinated subjects". The term absence might be too strong and this final statement should be mitigated. I think a proper conclusion should be: not detectable using our technical setting, or something of the kind.

Thank you for the suggestion. We have rephrased specifying that IgA2 (and not IgA in general) are not detected (see line 179).

Referee #3 (Comments on Novelty/Model System for Author):

The authors appear to have addressed all technical reservations expressed at first review and have expanded the manuscript acceptably.

Referee #3 (Remarks for Author):

The authors appear to have addressed all technical reservations expressed at first review and the conclusions drawn seem well supported by their data.

Minor issues

Line 111. Authors report "0.9374 +/- 0.02597" Can you really justify so many decimal places in the numbers reported? I am sure this is what the stats package reports but this is really 0.94 +/- 0.03 otherwise you are implying a precision of your measurements at 0.0001 which I doubt is the case.

We have corrected it.

Line 221 - Bad form to start sentences with Arabic numbers. Rephrase e.g. The secondary antibody mixture (50 uL) ...

We have changed it.

Throughout: I believe that OD is a defunct term and that most biochemical journals prefer absorbance e.g. A_{540nm} (number/unit as subscript)?

Thank you for the suggestion. We have replaced OD with absorbance (A_{405nm} and A_{450nm})(see Methods).

1st Apr 2022

Dear Maria,

Thank you for bearing with the last editorial issues. I am pleased to inform you that your manuscript is now accepted for publication!
As mentioned in my previous email, it will be published online as "just accepted".

Please read below for additional IMPORTANT information regarding your article, its publication and the production process.

Congratulations on your interesting work!

With my best wishes,

Lise

Lise Roth, Ph.D
Scientific Editor
EMBO Molecular Medicine

Follow us on Twitter @EmboMolMed
Sign up for eTOCs at embopress.org/alertsfeeds